# Parents of Adolescents Who Experience Suicidal Phenomena—A Scoping Review of Their Experience

**DOI:** 10.3390/ijerph20136227

**Published:** 2023-06-26

**Authors:** Demee Rheinberger, Fiona Shand, Lauren McGillivray, Sonia McCallum, Katherine Boydell

**Affiliations:** 1Black Dog Institute, University of New South Wales, Hospital Road, Randwick, NSW 2031, Australia; fionas@unsw.edu.au (F.S.); l.mcgillivray@blackdog.org.au (L.M.); k.boydell@blackdog.org.au (K.B.); 2Tyree Foundation Institute of Health Engineering, University of New South Wales, Sydney, NSW 2052, Australia; 3Centre for Mental Health Research, National Centre for Epidemiology and Population Health, The Australian National University, Canberra, ACT 2600, Australia; sonia.mccallum@anu.edu.au

**Keywords:** scoping review, suicide, self-harm, parents, adolescents

## Abstract

High prevalence rates of self-harm and suicide in adolescence provide unique challenges for parents. The aim of this scoping review was to explore key gaps in our understanding of the current scientific literature on the experience of parents who have adolescent children experiencing suicide crisis or self-harm. Four academic databases were searched using three broad concepts: self-harming behaviour or suicidal crisis; adolescents or young people; and the experiences or behaviour of parents, between journal inception and March 2022. Information reporting on the parents’ experience was extracted and a qualitative synthesis was conducted. Twenty-two articles met inclusion criteria and were assessed in detail. The experience of parents with an adolescent engaged in self-harm or suicidal crisis were classified into three temporal themes: discovery of the suicidal phenomena, management of suicidal phenomena, and after the suicidal phenomena had ceased. Parents caring for an adolescent experiencing self-harm or suicidal crisis experience poorer psychological wellbeing, difficulty accessing support services, and changes in the parent–child relationship. Parents desire greater support for both themselves and their child and further investigation is required to understand specifically which supports would be most appropriate at each stage.

## 1. Introduction

Suicide is the leading cause of death in Australian young people (aged 15–24 years) [1], with 34% of deaths in adolescents aged 15–17 years resulting from suicide [2]. Self-harm (self-inflicted injury with or without suicidal intent (the latter is also known as non-suicidal self-injury)) and suicide crisis (suicidal behaviours or ideation) peaks during adolescence [3,4], with estimates suggesting approximately 8% of Australian adolescents (aged 12–17 years) engage in self-harm [5] or experience suicidal thoughts [6] per year. Adolescents who have engaged in self-harm are also more likely to attempt suicide in the future [5,7], thereby considerably increasing their mortality risk [8].

Fortunately, positive adolescent-parent relationships have been shown to be the most consistent protective factor against adolescent self-harm with strong parental connections being associated with fewer instances of self-harm [9]. Parents are well placed to provide support outside of healthcare settings and can assist with means restriction, improve access to services, provide increased supervision to ensure safety [10], and encourage effective interpersonal connection, communication, and emotion regulation [11]. Furthermore, adolescents are more likely to tell a family member or friend about their self-harm incidents, than a mental health professional [12,13].

An adolescent engaging in self-harm or experiencing suicide crisis presents a unique challenge for parents, who are often ill-equipped to provide the emotional and psychological support their adolescent requires. A 2015 review of parents’ experiences of having an adolescent child engaging in non-suicidal self-injury (NSSI) indicated that parents experience numerous negative emotions, such as shock, sadness, and guilt, and experience an increase in mental ill-health in the period following an adolescent child’s self-harm incident [14]. A 2022 qualitative review of young peoples’ and caregivers’ experiences found that caregivers were unsure of how to support their child and had lost confidence in their parenting ability as a result of the suicide crisis or self-harm [15]. Since parents play a crucial role in ensuring the wellbeing of their adolescent children, it is vital to understand the unique experience of parents with adolescent children who are in a suicide crisis or engaging in self-harm so as to better understand how they can be supported. To date, no reviews have comprehensively examined both the qualitative and quantitative literature regarding parents’ experiences when they have an adolescent child engage in self-harm or a suicide crisis. As such, this scoping review aims to explore the breadth of empirical evidence of this experience and identify gaps in our current understanding. 

## 2. Materials and Methods

This scoping review protocol followed the Preferred Reporting Items for Systematic Review and Meta-Analyses extension for Scoping Reviews (PRISMA-ScR) framework [16] (checklist available in the Appendix A). This scoping review was conducted to understand the breadth of the existing scientific literature and key gaps in our understanding of the experience of parents who have adolescent children experiencing suicide crisis or self-harm. This broad research question allows for a wide exploration of this previously under-explored phenomenon. For the purposes of this review, suicide crisis (thoughts, plans, behaviours, attempts, or death) and self-harm (irrespective of suicide intent) will be collectively referred to as ‘suicidal phenomena’ [17] here forth.

### 2.1. Eligibility Criteria

A search was conducted for all studies exploring the experiences of parents of adolescents (aged 12–18 years), who had experienced a suicidal crisis (thoughts, plans, behaviours, attempts, or death) or self-harming behaviour (self-inflicted injury) without suicide intent. There was no restriction on publication date (up to the date of search), however studies were restricted to publications in the English language. Articles were excluded if they were not research studies (study protocols, editorials, theses, conference abstracts), were reviews (scoping or systematic reviews, meta-analyses, meta-syntheses), or explored experiences of primary carers, such as grandparents, foster parents, or residential carers. To account for broad age-ranges across studies that do not fit 12 to 18 years specifically, we included studies with participant ages ranging from 10 to 20 years if studies reported that more than 50% of the sample fell within 12 to 18 years.

### 2.2. Data Sources and Search Strategy

A search strategy was developed with assistance from the research team (DR, FS, KB, and LM) and a university librarian (MO) who had extensive experience undertaking literature searches within the mental health and medical fields.

The search was conducted via four databases: EMBASE, PsycINFO, CINAHL and PubMed. The search used three broad concepts: parents, adolescents, and suicide/self-harm, with both keywords and Medical Subject Headings (MESH) utilised when applicable. The search was initially conducted in EMBASE (Appendix B) before replicating the search strategy in the remaining three databases. Searches were conducted on the 9 March 2022.

All citations (*n* = 17,234) were imported into Endnote (Version 9), duplicates (*n* = 6397) were removed via Endnote’s automatic duplicate detection process, and a further 996 were removed after manual checking by the lead author. Citations were then taken from Endnote and imported into Covidence, a web-based screening and data extraction tool. Covidence automatically removed an additional 47 duplicates, resulting in a final sample of 9811 citations.

### 2.3. Screening

All screening was conducted via the Covidence platform. Two authors (DR and SM) initially screened a random sample of 10% (*n* = 981) to assess the appropriateness of the eligibility criteria and inter-screener agreement. Overall agreement between screeners was high (97.2%), suggesting appropriate eligibility criteria and consistency in screening. The remaining citations were then reviewed by the lead author, resulting in 377 articles eligible for full text screening. A random sample of 10% of full-text articles (*n* = 38) were screened by two authors (DR and LM), with an overall agreement of 81.6%. The remaining full text articles were screened by the lead author. Any disagreements between screeners (both during abstract/title and full text screening) were discussed between the two reviewers until consensus was reached before moving on to subsequent stages. Included for extraction were 38 articles, however an additional 16 were excluded during extraction due to further discussion amongst authors, leaving 22 articles in the final dataset. See Figure 1 for the PRISMA flow diagram.

### 2.4. Data Charting

A data charting template was developed by the research team based on the research question. It was iteratively updated throughout the extraction process as new areas of interest emerged. Data charting was conducted in Microsoft Excel by two members of the research team (DR and FS). The following information was extracted from each article and entered into the template:Descriptive information about the study: authors, title of publication, year of publication, country of origin, if the study was qualitative/quantitative/mixed methods, study aim, study design, focusing on suicide or self-harm (intent not disclosed) or NSSI, recruitment setting, age of child, child gender characteristics, participant population, sample size, gender proportions, and data source.Study details: Key findings, data collection method, sample details, analysis approach, findings in relation to parent’s experience, parent mental health, relationship with suicidal child, relationship with other immediate family, experiences of the child’s mental health, perceived reasons for adolescents’ suicide crisis/NSSI, managing suicidal crisis/NSSI behaviour, logistics of child’s care, and changes to lifestyle.

### 2.5. Data Summary and Synthesis

A reflexive thematic analysis [18] was conducted on the extracted data. Following the analysis phases outlined by Braun and Clarke [18], the first phase of analysis involved in-depth familiarisation by repeatedly reading through the data and making note of initial observations. As a result of familiarisation, the decision was made to temporally explore the data. As such, the data were grouped into distinct categories based on which point in the journey the parents experience had been explored. This resulted in three distinct time periods: (1) the discovery of the suicidal phenomena, (2) during the suicidal phenomena management, and (3) after the suicidal phenomena ceased. The second phase involved reviewing the data within each temporal category to determine which codes could be derived from the data. Codes were both deductive (arriving from within the data itself) and inductive (from broad understanding of the phenomena). These codes were then reviewed and grouped to create themes which were refined until the final set of sub-themes were determined. This analysis and synthesis were conducted by the lead author and was reviewed and refined in collaboration with KB.

Analytic rigour was maintained throughout the process via a team approach to analysis, maintenance of a detailed audit trail, ongoing reflexivity, and the authors having prolonged engagement with the topic area.

## 3. Results

Twenty-two articles met the inclusion criteria. They were published between 2000 and 2022, with 82% published since 2015 (inclusive). The majority of studies were qualitative (73%) and were conducted in the USA (*n* = 8, 36%), followed by China (*n* = 4, 18%), and Australia (*n* = 4, 18%). All studies reported on a unique participant group exploring parental experiences. In total, 2388 parents were included by studies in this review. Of the studies, 41% (*n* = 9) of studies specifically investigated suicide, 32% (*n* = 7) investigated self-harm irrespective of intent, and 27% (*n* = 6) investigated self-harm without suicide intent. Study characteristics are reported in Table 1.

Thematic analysis resulted in identification of three temporal themes, each with sub themes, which outlined the existing evidence as it pertains to the discovery of the suicidal phenomena, the management of suicidal phenomena, and after the suicidal phenomena had ceased (Table 2).

### 3.1. Discovery of Suicidal Phenomena

This theme relates to parents’ experiences upon the discovery of their adolescent child’s suicidal phenomena. This includes parents being unsure how to respond to the discovery and their emotional response.

#### 3.1.1. Unsure How to Respond

In many of the studies, parents were slow to respond to the suicidal phenomena [20,23,25,30,36,39]. Some parents reported choosing to ignore the behaviours [36,39], largely due to not understanding the severity of the suicidal phenomena [25,39] or misattributing the behaviour to accidents [39], attempts at manipulation [23], or as that typically purported to be part of puberty [39].

Some parents were resistant to implementing safety precautions [20], others reported that they did not know where to seek support for their child [30], or did not take support services seriously when these services raised concerns about the child’s suicidal phenomena [36].

#### 3.1.2. Emotional Response

Numerous studies reported strong emotional responses from parents when they discovered the suicidal phenomena [20,23,24,30,35,38,39,40]. Shock and surprise were typical for parents [23,30,35,40], alongside worry and anxiety about the situation and wellbeing for their child [30,35,38]. A retrospective quantitative study found that parents’ anxiety was highest on discovery but subsided somewhat the follow day [38]. The same study found that parents were more likely to be anxious if the adolescent had engaged in severe suicidal phenomena because they perceived their child being at a greater risk of death [38].

Some studies reported that parents responded to the discovery with anger and hostility [23,30,35,38,39]. Ye et al. [40] highlighted that parents were eager to understand the reasons for the suicidal phenomena, the uncertainty around which may have been contributing to the parents’ hostility.

Parents also expressed sadness as a result of the discovery. Wagner et al. [38] found that mothers were significantly more likely to experience sadness, rather than hostility on discovery, and O’Gara et al. [35] indicated that fathers who had a closer bond with their child were more likely to report feeling sadness on discovery. Parents viewed the discovery of the self-harm or suicide crisis as a traumatic event [40], while Ewell Foster et al. [24] found that 51.3% of parents experienced some psychological distress on discovery, however the majority of those experienced only mild symptoms.

### 3.2. During the Management of Suicidal Phenomena

This theme represents the parents’ experience during the adolescent’s suicidal phenomena. This theme contained the largest proportion of the evidence with data from 95% of articles included in the review. This theme explored parents’ wellbeing, parents’ efforts to engage in help-seeking for their adolescent child, the growing connection or disconnection between parent and child, the limited access to support for parents, changes in parents’ behaviours and the impact on their family, and parents’ attempts to make sense of the suicidal phenomena.

#### 3.2.1. Parent Wellbeing

Of studies which explored parents’ experience during the management of suicidal phenomena, eighty-one percent indicated that managing the adolescent’s suicidal phenomena had a significant impact on their wellbeing [20,21,22,23,24,25,27,28,29,30,32,35,36,37,38,39,40]. Physiological impacts on parents were common, with parents reporting reduced functional capacity, changes in their sleep behaviours (e.g., insomnia), and nausea [24,25,29,30,37]. Overwhelmingly, parents reported psychological impacts, with 66.7% to 81% [30,37] of parents reporting some negative psychological symptoms including anxiety, stress, depression, and grief [20,24,27,28,30,36,37]. Parents also reported experiencing rumination, intrusive thoughts [37], and emotion dysregulation [20]. Anxiety and worry were particularly pronounced in parents as they attempted to manage the suicidal phenomena [19,24,36,37,38]. This anxiety appeared to originate from two concerns, firstly that the child would not receive adequate help and the suicidal phenomena would reoccur or they would lose their lives [25,29,32], and secondly from worry that they had been responsible for the suicidal phenomena [22,23,37,39,40].

Parents reported being tormented by the idea that they had been responsible for their adolescent’s crisis [22,23]. Self-blame or feelings of guilt were reported in just under a third of the studies within this subtheme [22,23,37,39,40], with parents noting their own high expectations, divorce, or family history of mental health problems as possible precipitants that they were responsible for [39]. Guilt was also experienced due to parents’ belief they had failed to protect their child and keep them safe [23,25,30,36,37,40]. One study exploring parents’ self-efficacy to maintain their child’s safety from future suicidal phenomena found that parents tended to evaluate themselves as unable to do so, with more severe instances of self-harm, or more prolonged crises resulting in lower ratings of confidence to keep their child safe [21].

#### 3.2.2. Help-Seeking for Child

Of articles exploring parents’ experiences during the management of the adolescent’s suicidal phenomena, 52% discussed parent’s attempts to seek help for their child. Of those that did report seeking help, most focused on the experiences of seeking help from school personnel [20,30], hospitals [23,24,26,32], and mental health professionals in the community [24,30]. However, not all parents were eager to engage in help-seeking for their child, with Kelada et al. [30] finding only half of the parents surveyed reported their child had contact with a mental health professional for their suicidal phenomena. Hesitance of parents to seek help for their child arose from concerns that a mental health professional would not be helpful [30], preferring to wait for the crisis to resolve itself [36], and fear of stigma or repercussions such as impacts on future education or employment options [25,35]. Parents also reported difficulty navigating the system and accessing services [26,32,40].

When engaged with support services parents reported a mix of experiences. Negative experiences included feeling as though support services did not prioritise psychological care for the suicidal phenomena [26], stigma from healthcare professionals [30], or inadequate support options [26]. Alternatively, positive experiences included proactive and knowledgeable staff [32], and access to peer workers [32]. Ewell Foster et al. [24] found that those who were fairly accepting of treatment were more likely to implement means restriction.

A few studies also outlined the types of support that parents would like to have received from services, including how to care for the child during future suicidal phenomena [23,31], greater access to in-patient care [24,32], medication prescriptions or adjustments for the adolescent [24], more in depth, adolescent-specific psychotherapy in the hospital [26,32] and in the community [32], easier access to peer support [32], and access to mental health education resources [24].

#### 3.2.3. Parent and Child (Dis)connection

This subtheme explores the proximity of the parent–child relationship as they attempt to navigate the management of the suicidal phenomena. Tense and fragile relationships were reported by Kelada et al. [30] and O’Gara et al. [35]. Czyz et al. [21] found that parents’ trust in the adolescent eroded with more instances of suicidal phenomena and Kaufman et al. [28] found that parent–child dyads engaged in more conflict if the child had engaged in suicidal phenomena than if they had not. Punishment for, and conflict around, the suicidal phenomena negatively impacted parent–child relationships [19,30], as did parents avoiding conversation or not acknowledging the suicidal phenomena [22,25,30]. Even when suicidal phenomena was acknowledged, some studies showed that parents did not realise how serious the suicidal phenomena was and the role that declining mental health played [25,35]. In many instances, studies reported that parents struggled to relate to their adolescent, specifically in relation to the suicidal phenomena [20,22,35,36,39,40]. Despite this, parent and child connections were fostered when parents increased their communication with their child [19,20,21,34,39,40] and improved their communication style, such as increasing validation [19,20,30,38] and compromise [20,22,25,30].

#### 3.2.4. Poor Access to Support for Parents

While studies indicated that parents sought support from a variety of services including mental health professionals, hospital staff, school mental health wellbeing staff, and peer support, numerous studies reported the challenges parents had accessing support during their child’s suicidal phenomena. Parents reported wanting more information from professionals around how to communicate with, manage, and support their child during this difficult time, however many did not receive that information from healthcare professionals [23,26,32]. Additionally, de Miranda Trinco et al. [23], Kelada et al. [30], and McKay and Shand [32] reported that parents wanted more information about the implications of, and recovery from, the suicidal phenomena but did not receive that information from health professionals.

Kelada et al. [31] found that parents wanted professional help to manage their own distress. Studies reported mixed outcomes for parents who did have access to mental health professionals. Kelada et al. [30] found 75% of parents who had access to a mental health professional reported a negative experience. Furthermore, parent’s also experienced stigma and judgement from health professionals when their child was hospitalised due to the suicidal phenomena [23]. However, other studies indicated that parents found interactions with mental health professionals to be helpful [31,32,36]. Helpful interactions included consistent, supportive communication [31,32] and services which informed and educated parents how to manage their emotional wellbeing [36]. Despite some positive experiences, one study reported that participants found support services difficult to access [31], and Townsend et al. [37] reported worse mental health outcomes for individuals not engaged with support.

Three studies found that the parents’ informal supports (i.e., family and friends) decreased following their child’s suicidal phenomena [22,25,37], with reasons including stigma and shame sharing the incident/s with their social group [22,25] or fear of leaving their child alone [25,37].

#### 3.2.5. Changes in Parent Behaviour and Impact on Family

Parents made considerable changes to their behaviour in response to adolescents’ self-harm or suicide crisis. Most studies which outlined changes in parent behaviour identified increases in parental vigilance of the adolescent [22,30,36,37,39,40]. In some instances, this involved sleeping next to the child to ensure their safety throughout the night [39,40]. One study also reported that parents were monitoring their child in an attempt to immediately mediate any emotional distress or potential triggers [37]. To accommodate this increased vigilance and monitoring, parents reduced work commitments, either reducing the number of hours spent in paid employment or leaving employment entirely, which created or exacerbated financial stress [25,37].

Changes in parent behaviour also had an impact on the wider family, with numerous studies reporting changes in family dynamics. In many instances, parents attributed the change of family dynamics to reduced communication and increased relationship strain [22,36,37]. This appeared to negatively impact the family as parents struggled to meet the needs of their other children [36,37] and put strain on spousal relationships [37]. In addition, the suicidal phenomena were shown to have a negative impact on siblings [37], as they attempted to protect the sibling who engaged in suicidal phenomena or were relatively neglected by parents as a result of parents attempting to manage the suicidal phenomena [22,36,37].

#### 3.2.6. Making Sense of the Suicidal Phenomena

In response to managing the suicidal phenomena, parents reported attempting to understand why it was occurring and what purpose it had for the adolescent. Oldershaw et al. [36] and Wang et al. [39] demonstrated that parents’ understanding of the crisis was that their adolescents’ were attempting to cope with strong, negative emotions. Ironically Wang et al. [39] noted the perceived role that the parents played in eliciting the strong, negative emotions that prefaced self-harm.

Despite some parents struggling to make sense of the crisis [39], Ewell Foster et al. [24] found no association between parental stigma and the likelihood of them implementing discharge recommendations. Czyz et al. [21] found that parents generally reported high self-efficacy for suicide prevention activities, and that greater lifetime history of suicide attempt increased parents self-rated efficacy to ask about mood and suicidal thoughts, and that increased adolescent age was associated with greater self-rated ability to notice warning signs.

### 3.3. After the Suicidal Phenomena Ceased

This theme relates to parents’ experiences after the suicidal phenomena has ended, be that through progression to long term management, without imminent risk of engaging in suicidal phenomena again, or the parents are bereaved.

#### 3.3.1. Changes in Parent–Child Relationship

Studies indicated there were noticeable changes in the parent–child relationship after having navigated the suicidal phenomena. Some studies noted that the relationship between parent and adolescent had deteriorated as a result of the suicidal phenomena [22,30]. However, the opposite was also seen within and between studies, with some finding that the relationship had improved after the suicidal phenomena due to improved trust, communication, and positive changes in family dynamics, such as more democratic decision making [30,36,39].

#### 3.3.2. Support for Parents

Parents reported wanting more support for themselves once their child’s treatment had ended [30] or after they became bereaved due to the suicide of their child [33], which did not differ from the support mentioned previously (Section 3.2.4). Furthermore, bereaved parents wanted peer support in the aftermath of the death, professional support to facilitate access to referrals, and other avenues for informal and formal supports [33].

## 4. Discussion

This scoping review sought to understand the breadth of the existing scientific literature and identify gaps in the existing understanding of the experience of parents who have adolescent children in suicide crisis or experiencing self-harm, to better understand how parents can be supported. Findings from this review of 22 articles support prior reviews that show parent wellbeing and mental health is adversely impacted when they have an adolescent child who experiences suicidal phenomena. This study extends this body of research by identifying the ways in which parents struggle when they have an adolescent child engage in suicidal phenomena and, more specifically, details the support that parents desire for both themselves and their child.

Having an adolescent child engage in suicidal phenomena has a profound impact on parents’ psychological wellbeing, with increased psychological distress including anxiety, depression, and guilt. While anxiety was a common emotional response on discovery of the suicidal phenomena [30,35,38], anxiety and worry persisted throughout the parent experience [20,24,36,37,38] as parents lived with fear that their adolescent child would die by suicide. Physiological changes were also common on discovery and during the management of the suicidal phenomena, such as sleep disturbances [24,25,29,30,37]. Evidence has shown that poor sleep quality is linked to poorer mental health [41], which could compound the impacts of already eroding parent mental wellbeing due to caring for an adolescent in crisis. Poor mental wellbeing has been reported in other reviews, finding that parents of adolescents engaging in suicidal phenomena report high levels of anxiety, sadness, and guilt [14,15].

Despite the considerable mental health implications for parents, many studies identified that parents had insufficient access to support services [23,26,32]. Parents reported wanting more support from mental health professionals which was free of stigma [23] and provided constructive information about how to engage with and manage their adolescent and the crisis [23,26,32], as well as support to improve their own mental health [31]. However, while evidence indicated that utilising mental health support led to better mental health outcomes [37], parents report that this support is difficult to access [31]. Access to professional mental health support is consistently impacted by high fees, long waiting times, and time-consuming referral pathways [42,43]. While parents may recognise that this support is necessary, they may choose to prioritise access to care for their adolescent, leaving no time or financial resources left to access care for themselves. To further exacerbate this issue, informal support such as family and friends, often deteriorate [22,25,37] as parents find it difficult to maintain relationships due to changing priorities, fear of stigma, and increased vigilance of their adolescent. While informal supports may not provide therapeutic support, a strong social network is important for ensuring good mental wellbeing [44] and the absence of these informal supports may further impact parents’ wellbeing during this experience.

Parents’ attempts at making sense of the suicidal phenomena was identified during the management of the suicidal phenomena. Sense-making is a central theme in studies of chronic illness, another time when parents are faced with no definitive end date for the child’s struggle. Sense-making is a process by which individuals try to understand adverse events in their life by attempting to fit them into their extant everyday ways of viewing and experiencing the world; this can range from rumination and the question of why this has happened to seeking out information and sharing sense-making assumptions with others [45]. A 2017 study with 41 parents found that most parents were bewildered by self-harm in their children, and that making sense of the self-harm took considerable effort and was emotionally taxing [46]. However, sense-making provided parents with new tools to help them understand their adolescent and facilitated their navigation of their new worldview and experience [46]. Therefore, future support options for parents who have an adolescent child experiencing suicidal phenomena may benefit from being provided activities which facilitate and encourage sense-making.

The current findings indicated that parents require a higher degree of support to care for themselves and their child. Support typically available for parents to date has included access to mental health professionals and general practitioners (GPs). However, parents indicated that this support was sometimes unhelpful or insufficient [23,30]. Temporal analysis showed that parents require support during the management of the suicidal phenomena and once the suicidal phenomena has ceased. Future research should aim to understand specifically what support parents need at each timepoint, and when is an appropriate time to deliver that support.

The absence of parent-reported need for support during discovery may be a result of low rates of parent acknowledgment of the seriousness of the suicidal phenomena at this time. As such, parents may benefit from more education about the risks associated with suicidal phenomena and appropriate responses to discovery. A recent Australian study exploring parents of young people who self-harm found that parents wanted more information about what self-harm is, and what the risks were [47]. Interestingly, while our review found that some parents were engaging in help-seeking for themselves and their adolescent, no studies reported on education and information seeking from non-professional sources, such as accessing information online. It is unclear if and how parents engage in this form of help-seeking and what gaps and benefits it may have. Additionally, more investigation is required to understand what support parents may find useful throughout the suicidal phenomena experience. Parents of children with traumatic brain injury, an experience similar to that of parents in this review [48], report finding peer support in the initial stages beneficial, as it provided a sense of connection, important information about their role in the child’s care going forward, and emotional compassion and understanding [48]. Peer support is a growing area of interest for suicide prevention [49], however, no studies have examined its use for parents or carers of those experiencing suicidal phenomena. Future research should examine if peer support for parents during the discovery is beneficial and if it has positive impacts of parent wellbeing throughout the suicidal phenomena experience.

While after the cessation of the suicidal phenomena was determined to be a distinctive point in parents’ experience, there was limited research exploring what this time is like for parents. Participants’ needs and experiences after cessation are likely to be unique as they move from a path of active surveillance to fear of reoccurrence, ongoing mental health management, or learning how to navigate life as a bereaved parent. Parents bereaved by the suicide of offspring of all ages have reported that primary care such GPs were an important avenue for support [50]; however, GPs are often unsure how to support suicidally bereaved parents [51]. Unfortunately, while formal support options are likely to assist bereaved parents [52,53], there is insufficient evidence to confirm that existing interventions are actually beneficial for bereaved parents [52]. More research needs to be conducted to understand parents’ experiences after the cessation of the suicidal phenomena and how parents can be better supported long term.

Parents of adolescents experiencing suicidal phenomena are more likely to have a history of mental health problems themselves [54], suggesting that these parents may experience greater psychological distress than parent–child dyads without any family history of mental health problems. Furthermore, there is evidence that these parents, particularly mothers, are likely to have a history of suicide attempts themselves [55]. As such, any support provided to parents may need to be designed to account for a heightened level of baseline distress or lower functioning. Future research may be necessary to understand if this higher degree of psychological distress accounts for the low rates of effectiveness which parents have reported from mental health professionals [23,30] and whether mentally healthy parents require the same support as those with existing poor mental health. Future research should also explore if having a personal history of experiencing suicidal phenomena influences parents’ response to their adolescent’s suicidal phenomena (e.g., influencing help-seeking behaviours or how they respond to their adolescent).

### 4.1. Limitations of the Papers Reviewed

Most studies involved mothers and very few to no fathers, as a result little is known about the experience of fathers when they have an adolescent experiencing suicidal phenomena. Fathers may have differing emotional and interpersonal responses to the suicidal phenomena and require different types of support. Additionally, most studies recruited participants from clinical or hospital settings, suggesting a greater severity of adolescent’s suicidal phenomena (i.e., requiring medical intervention), which resulted in the adolescent accessing these services. This would likely affect parents’ experiences of the adolescents’ suicidal phenomena, compared to less severe instances of suicidal phenomena. Furthermore, a considerable proportion of the studies recruited parents from emergency department or hospital settings, but very few studies explored the experience of using these emergency services. Given that parents are likely to be in a higher degree of distress if they have sought urgent care for their child, the experiences and needs of parents might differ. Finally, most studies were conducted in upper-middle- or high-income countries and little is known about the experience and support needs of parents in low-middle-income countries.

### 4.2. Limitations of This Review

There are a few limitations to this scoping review. First, we excluded articles that were not in English, were review papers, or were not peer reviewed. Second, while the journal databases searched for this review were those commonly used for social science and medical journals, it is possible that articles exploring experience were not in those databases, and therefore were missed. Third, “experience” has a broad definition, and only one reviewer (DR) conducted most of the full text screening, so selected articles may have been influenced by researcher bias or personal understanding of the term. However, given the high degree of agreement between researchers at both screening stages, it is unlikely this had a significant impact on the articles included in this review. Finally, there is a considerable time delay between the search being conducted and the manuscript being finalised and more current research which may have provided additional support or novel ideas to this review which were not included.

## 5. Conclusions

The experience of parents who are caring for an adolescent child experiencing suicidal phenomena is often fraught with challenges. Three temporal themes were identified; the discovery of the suicidal phenomena, during the management of the phenomena, and after the suicidal phenomena has ceased. This scoping review found that the experience impacted parents’ mental health, interpersonal relationships, daily functioning, and relationship with their adolescent from discovery through to the cessation of the suicidal phenomena. Parents, who are often overwhelmed with anxiety and fear of future incidents or death, typically want more access to support for their child and themselves. Despite the difficult situation, some parents found that the experience improved the relationships between themselves and their adolescent, and the immediate family. This review indicates that more investigation is required to understand specifically which supports would be most appropriate for parents at each stage of the suicidal phenomena experience.

## Figures and Tables

**Figure 1 ijerph-20-06227-f001:**
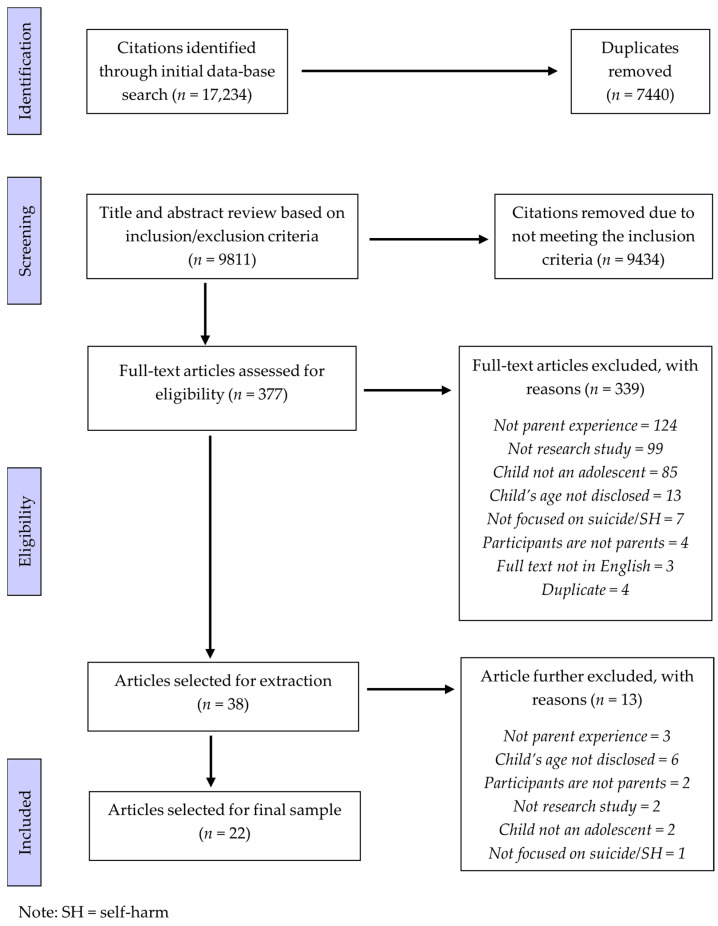
PRISMA Flow Diagram.

**Table 1 ijerph-20-06227-t001:** Characteristics of studies.

Authors and Year	Country	S/NSSI/SH	Participants	Adolescent Demographics	Data Source	Recruitment Method	Temporal Theme/s
Baetens et al. (2015) [19]	Belgium	NSSI	1438 parents (88.7% female) *	12 years old, 54.7% female	Self-report surveys	Not reported	During management
Berk et al. (2022) [20]	USA	Suicide attempt and NSSI	12 parents (75% female) Age 45–56 (mean age 49.4 years)	13–17-year-olds (mean age 14.7 years), 70% female	Self-report surveys and structured interviews	Recruited through psychiatric clinic	Discovery, During management
Czyz et al. (2018) [21]	USA	Suicide	162 parents (79.6% female)	13–17 years (mean age = 15.41 years), 57% female	Self-report surveys	Recruited though psychiatric emergency department services	During management
Daly (2005) [22]	Canada	Suicide	6 parents (100% female) Age range = 32–45 years	12–16 years	Unstructured interviews	Outpatient family therapy provider	During management, After
de Miranda Trinco et al. (2017) [23]	Portugal	NSSI	38 parents (89% female)	13–18 years	Semi-structured interviews	Emergency department in a paediatric hospital	Discovery, During management
Ewell Foster et al. (2021) [24]	USA	Suicide	118 parents (73.7% female)	11–18 years, 57.3% female	Self-report surveys (baseline and follow-up)	Emergency department	Discovery, During management
Fu et al. (2020) [25]	China	NSSI	20 parents (80% female)	12–18 years (mean age = 14.5 years) 85% female	Semi-structured interviews	Child psychiatric ward	Discovery, During management, After
Fu et al. (2021) [26]	China	Suicide and self-harm	15 parents (73% female)	12–18 years (mean age = 14.2 years)	Semi-structured interviews	Psychiatric department in a general hospital	During management
Gillespie et al. (2019) [27]	Ireland	Self-harm	167 parents (72.5% female) Age range = 25–65+	13–18 years	Self-report surveys	Community based child and adolescent mental health service	During management
Kaufman et al. (2020) [28]	USA	Self-harm	60 parents (30 control and 30 of self-injuring adolescents) (100% female) Mean age clinical sample = 45.07 years Mean age control sample = 44.8 years	13–17 years (mean age clinical sample = 15.47 years, mean age control sample = 14.77 years) 100% Female	Interview and self-report surveys	Online recruitment, paediatrician offices and local businesses, and outpatient and inpatient clinics	During management
Kawabe et al. (2016) [29]	Japan	Suicidal ideation	179 parents and 6 caregivers (83% female) Median age 43 years	12–15 years 51.4% female	Self-report surveys	3 junior high schools in rural region of Japan	During management
Kelada et al. (2016) [30]	Australia	NSSI	16 parents (94% female) 36–56 years (mean age = 45.44 years)	14–17 years (mean age = 15.38 years) 62.5% female	Self-report surveys	5 high schools	Discovery, During management, After
Kelada et al. (2017) [31]	Australia	NSSI	10 parents (100% female) Mean age = 45.2 years	Mean age = 15.1 years 90% female	Survey with open ended questions	High schools	During management
McKay & Shand (2016) [32]	Australia	Suicide attempt	3 parents (100% female)	14–15 years 100% female	Semi-structured interviews	Online recruitment	During management, After
Miers et al. (2012) [33]	USA	Suicide death	8 parents (75% female) Age range = 33–50 years	13–18 years 33.3% female	Semi-structured interviews	Suicide bereavement groups	After
O’Brien et al. (2019) [34]	USA	Suicide	8 parents (50% female) Mean age = 53.5 years	Mean age = 16.5 years 75% female	Semi-structured interviews	Inpatient psychiatric unit	During management
O’Gara et al. (2022) [35]	USA	Suicide	10 fathers Mean age = 39.4 years	Age range = 11–19 years Mean age = 15.9 years 100% female	Semi-structured interviews	Mental health services associated with 3 hospitals and outpatient departments	Discovery, During management
Oldershaw et al. (2008) [36]	England	Self-harm	12 parents (83% female)	Age range = 13–18 years 100% female	Semi-structured interviews	Community child and adolescent mental health services	Discovery, During management, After
Townsend et al. (2021) [37]	Australia	Self-harm	37 parents (92% female) Mean age = 45.7 years	Age range = 12–18 years Mean age = 16.89 years 75.7% female	Self-report survey with open ended questions	Online recruitment and via community mental health services and parenting groups	During management
Wagner et al. (2000) [38]	USA	Suicide attempt	34 parents (65% female) Mothers age range = 32–49 years (median = 39 years) Fathers age range = 38–55 years (median = 46 years)	Age range = 13–19 years Mean age = 15.5 years	Self-report survey	Private psychiatric hospital	Discovery, During management
Wang et al. (2022) [39]	China	NSSI	24 parents (75% female) Mean age = 42.5 years	Age range = 12–18 years Mean age = 15 years 87.5% female	Semi-structured interviews	Psychiatric ward	Discovery, During management, After
Ye et al. (2021) [40]	China	Self-harm	11 parents (73% female) Age range = 35–40 years Mean age = 37 years	Age range = 11–18 years	Semi-structured interviews	Emergency department	Discovery, During management

NOTE: S = suicide, NSSI = non-suicidal self-injury, SH = self-harm with intent not disclosed * at time point 1.

**Table 2 ijerph-20-06227-t002:** Temporal themes and sub-themes.

	Temporal Themes	Sub-Themes
1.	Discovery of suicidal phenomena	1.1 Unsure how to respond
		1.2 Emotional response
2.	During the management of suicidal phenomena	2.1 Parent wellbeing
		2.2 Help seeking for child
		2.3 Parent and child (dis)connection
		2.4 Poor access to support for parents
		2.5 Changes in parent behaviour and impact on family
		2.6 Making sense of the crisis
3.	After the suicidal phenomena had ceased	3.1 Changes in parent–child relationship
		3.2 Support for parents

## Data Availability

No new data was created.

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
