# Peer review of "Parents of Adolescents Who Experience Suicidal Phenomena—A Scoping Review of Their Experience"

_ijerph, 2023, doi:10.3390/ijerph20136227_

Round 1

Reviewer 1 Report

The review addresses the important topic of the experience of parents of adolescents who self-harm. Overall, the review has been well conducted, and the manuscript has been well-written and clearly structured. Nonetheless, I have a few methodological concerns and questions for clarifications.

Introduction

Introduction is brief and to-the-point. However, it would be helpful if there was a definition of ‘suicide crisis’ and ‘self-harm’.

Authors also refer to NSSI. Is that the same? (it was included in data extraction).

Methods

The review included “studies with participant ages ranging from 10 to 20 years if more than 50% of the sample fell within 12 to 18 years.”

Please clarify how authors assessed ages of participants in selected studies.

Also, several studies in the table say: “Age not reported”. I would expect that these studies would have been excluded.

Searches have been conducted in March 2022. It is a good practice that searches are within 6 months before submission. Please update the searches.

Inter-rater agreement: please report kappa coefficient instead of percentage.

It is common that one author conducts the title/abstract screening. However, full-text screening should be done by two authors. It is unclear why only 10% was done by two authors, and it should be mentioned with the study limitations.

Out of curiosity: Why do you call the themes ‘temporal’ themes?

Check for typos throughout.

Check for typos

Author Response

Please see the attachment for response to both reviewers. 

Reviewer 2 Report

Thank you for submitting your well written work on the experience of parents with adolescents who self-harm or think about suicide. Your work has given me more knowledge in this field of suicide-research and it was a pleasure reviewing your manuscript. Your work is suitable for publication and your analyses can further support young people and their parents who may experience a suicidal crisis. I have minor suggestions below that may help to improve parts of your work.

Title: Although the title is proper, after reading the manuscript I would suggest for the authors to use only suicidal phenomena in the title or just use suicidal crisis and omit self-harm.

Introduction: Well written. To my understanding the authors in line 31 refer to self-harm as the self-harming behaviour, such as cutting ‘Adolescents who have engaged in self-harm are also more likely to attempt suicide in the future’. However, we have issues internationally on the suicide terminology nomenclature and often when we use ‘self-harm’ as a term we refer to a suicide attempt with and without suicidal intent. Therefore, I would suggest to the authors to clarify from the first paragraph what they mean with the term self-harm and be consistent throughout the paper. I think this will help readers.

Discussion:

·         Page 13, 3rd para: ‘The focus on a greater need for support during the management of the suicidal phenomena and after the suicidal phenomena has ceased rather than on discovery may be a result of low rates of parent acknowledgment of the seriousness of the suicidal phenomena at discovery’. This sentence is quite long with no commas anywhere and makes it a bit hard to read. Could the authors edit this and make it easier to understand?

·         Page 14, 2nd para: ‘Parents of adolescents…mental health’. The authors make a very important point here about the history of mental health conditions of parents. Clinically we also know that in some cases of adolescents who engage with self-harming, their parents further had similar experiences with self-harming or attempts or ideation. There are some older studies on this, i.e. https://doi.org/10.1016/j.jaac.2013.12.022. Could the authors add few lines on this association and maybe suggest this for future studies?

Author Response

Please see the attachment for response to both reviewers
